# HOW DOES LEARNING RATE DECAY HELP MODERN NEURAL NETWORKS?

## ABSTRACT

Learning rate decay (lrDecay) is a *de facto* technique for training modern neural networks. It starts with a large learning rate and then decays it multiple times. It is empirically observed to help both optimization and generalization. Common beliefs in how lrDecay works come from the optimization analysis of (Stochastic) Gradient Descent: 1) an initially large learning rate accelerates training or helps the network escape spurious local minima; 2) decaying the learning rate helps the network converge to a local minimum and avoid oscillation. Despite the popularity of these common beliefs, experiments suggest that they are insufficient in explaining the general effectiveness of lrDecay in training modern neural networks that are deep, wide, and nonconvex. We provide another novel explanation: an initially large learning rate suppresses the network from memorizing noisy data while decaying the learning rate improves the learning of complex patterns. The proposed explanation is validated on a carefully-constructed dataset with tractable pattern complexity. And its implication, that additional patterns learned in later stages of lrDecay are more complex and thus less transferable, is justified in real-world datasets. We believe that this alternative explanation will shed light into the design of better training strategies for modern neural networks.

## 1 INTRODUCTION

Modern neural networks are deep, wide, and nonconvex. They are powerful tools for representation learning and serve as core components of deep learning systems. They are top-performing models in language translation (Sutskever et al., 2014), visual recognition (He et al., 2016), and decision making (Silver et al., 2018). However, the understanding of modern neural networks is way behind their broad applications. A series of pioneering works (Zhang et al., 2017; Belkin et al., 2019; Locatello et al., 2019) reveal the difficulty of applying conventional machine learning wisdom to deep learning. A better understanding of deep learning is a major mission in the AI field.

One obstacle in the way of understanding deep learning is the existence of magic modules in modern neural networks and magic tricks to train them. Take batch normalization module (Ioffe & Szegedy, 2015) for example, its pervasiveness in both academia and industry is undoubted. The exact reason why it expedites training and helps generalization, however, remains mysterious and is actively studied in recent years (Bjorck et al., 2018; Santurkar et al., 2018; Kohler et al., 2019). Only when we clearly understand these magical practices can we promote the theoretical understanding of modern neural networks.

Learning rate is "the single most important hyper-parameter" (Bengio, 2012) in training neural networks. Learning rate decay (lrDecay) is a *de facto* technique for training modern neural networks, where we adopt an initially large learning rate and then decay it by a certain factor after pre-defined epochs. Popular deep networks such as ResNet (He et al., 2016), DenseNet (Huang et al., 2017b) are all trained by Stochastic Gradient Descent (SGD) with lrDecay. Figure 1(a) is an example of lrDecay, with the learning rate decayed by 10 every 30 epochs. The training is divided into several stages by the moments of decay. These stages can be easily identified in learning curves (such as Figure 1(b)), where the performance boosts sharply shortly after the learning rate is decayed. The lrDecay enjoys great popularity due to its simplicity and general effectiveness.

Common beliefs in how lrDecay works are derived from the optimization analysis in (Stochastic) Gradient Descent (LeCun et al., 1991; Kleinberg et al., 2018). They attribute the effect of an initially

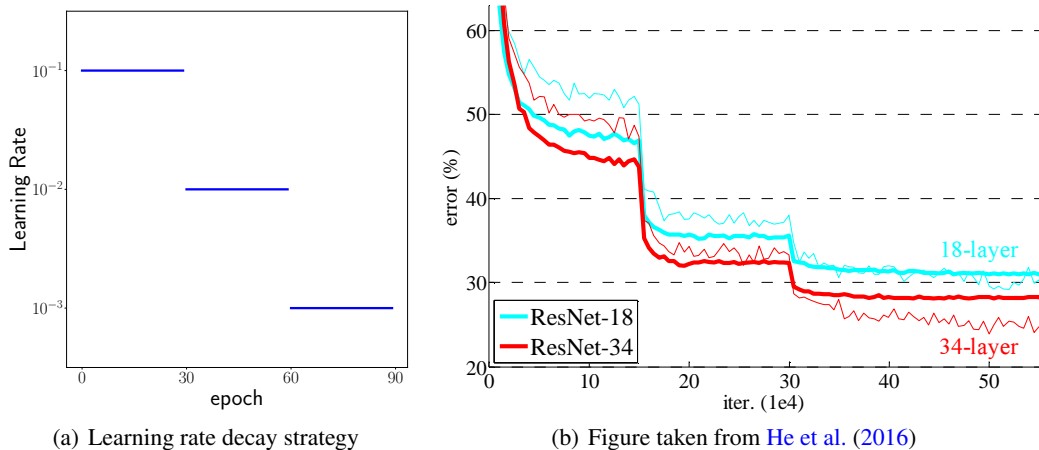

(a) Learning rate decay strategy      (b) Figure taken from He et al. (2016)

Figure 1: Training error in (b) is shown by thin curves, while test error in (b) by bold curves.

| explanation | perspective | initially large lr | supported | lr decay | supported |
|---|---|---|---|---|---|
| LeCun et al. (1991) | optimization | accelerates training | ✓ | avoids oscillation | ✗ |
| Kleinberg et al. (2018) | optimization | escapes bad local minima | ✓ | converges to local minimum | ✗ |
| Proposed | pattern complexity | avoids fitting noisy data | ✓ | learns more complex patterns | ✓ |

Table 1: Comparison of explanations on why lrDecay helps training neural networks. The column "supported" means whether the explanation is supported by the empirical experiments in this paper.

large learning rate to escaping spurious local minima or accelerating training and attribute the effect of decaying the learning rate to avoiding oscillation around local minima. However, these common beliefs are insufficient to explain our empirical observations from a series of carefully-designed experiments in Section 4.

In this paper, we provide an alternative view: the magnitude of the learning rate is closely related to the complexity of learnable patterns. From this perspective, we propose a novel explanation for the efficacy of lrDecay: **an initially large learning rate suppresses the memorization of noisy data while decaying the learning rate improves the learning of complex patterns.** This is validated on a carefully-constructed dataset with tractable pattern complexity. The pattern complexity in real-world datasets is often intractable. We thus validate the explanation by testing its implication on real-world datasets. The implication that additional patterns learned in later stages of lrDecay are more complex and thus less transferable across different datasets, is also justified empirically. A comparison between the proposed explanation and the common beliefs is summarized in Table 1. Our explanation is supported by carefully-designed experiments and provides a new perspective on analyzing learning rate decay.

The contribution of this paper is two-fold:

- We demonstrate by experiments that existing explanations of how lrDecay works are insufficient in explaining the training behaviors in modern neural networks.
- We propose a novel explanation based on pattern complexity, which is validated on a dataset with tractable pattern complexity, and its implication is validated on real-world datasets.

The explanation also suggests that complex patterns are only learnable after learning rate decay. Thus, when the model learns all simple patterns, but the epoch to decay has not reached, immediately decaying the learning rate will not hurt the performance. This implication is validated in Section A.1.

## 2 RELATED WORK

### 2.1 UNDERSTANDING THE BEHAVIOR OF SGD

Recently, researchers reveal the behavior of SGD from multiple perspectives (Li et al., 2019; Mangalam & Prabhu, 2019; Nakkiran et al., 2019). They respect the difference among data items rather

than treat them as identical samples from a distribution. They study the behavior of SGD in a given dataset. In Mangalam & Prabhu (2019), they show that deep models first learn easy examples classifiable by shallow methods. The mutual information between deep models and linear models is measured in Nakkiran et al. (2019), which suggests deep models first learn data explainable by linear models. Note that they are not relevant to learning rates. Li et al. (2019) analyze a toy problem to uncover the regularization effect of an initially large learning rate. Their theoretical explanation is, however, based on a specific two-layer neural network they design. Different from these works, Section 5 studies the behavior of SGD induced by lrDecay in a modern WideResNet (Zagoruyko & Komodakis, 2016), finding that learning rate decay improves learning of complex patterns. We formally define pattern complexity by expected class conditional entropy, while the measure of pattern complexity in Mangalam & Prabhu (2019); Nakkiran et al. (2019) relies on an auxiliary model.

## 2.2 ADAPTIVE LEARNING RATE METHODS

Adaptive learning rate methods such as AdaGrad (Duchi et al., 2011), AdaDelta (Zeiler, 2012), and ADAM (Kingma & Ba, 2015) are sophisticated optimization algorithms for training modern neural networks. It remains an active research field to study their behaviors and underlying mechanisms (Reddi et al., 2018; Luo et al., 2019). However, we focus on learning rate decay in SGD rather than on the adaptive learning rate methods. On the one hand, SGD is the *de facto* training algorithm for popular models (He et al., 2016; Huang et al., 2017b) while lrDecay is not common in the adaptive methods; On the other hand, many adaptive methods are not as simple as SGD and even degenerate in convergence in some scenarios (Wilson et al., 2017; Liu et al., 2019). We choose to study SGD with lrDecay, without introducing adaptive learning rate to keep away from its confounding factors.

## 2.3 OTHER LEARNING RATE STRATEGIES

Besides the commonly used lrDecay, there are other learning rate strategies. Smith (2017) proposes a cyclic strategy, claiming to dismiss the need for tuning learning rates. Warm restart of learning rate is explored in Loshchilov & Hutter (2017). They achieve better results when combined with Snapshot Ensemble (Huang et al., 2017a). These learning rate strategies often yield better results at the cost of additional hyperparameters that are not intuitive. Consequently, it is still the *de facto* to decay the learning rate after pre-defined epochs as in Figure 1(a). We stick our analysis to lrDecay rather than to other fancy ones because of its simplicity and general effectiveness.

## 2.4 TRANSFERABILITY OF DEEP MODELS

Training a model on one dataset that can be transferred to other datasets has long been a goal of AI researches. The exploration of model transferability has attracted extensive attention. In Oquab et al. (2014), deep features trained for classification are transferred to improve object detection successfully. Yosinski et al. (2014) study the transferability of different modules in pre-trained networks, indicating that higher layers are more task-specific and less transferable across datasets. By varying network architectures, Kornblith et al. (2019) show architectures with a better ImageNet accuracy generally transfer better. Raghu et al. (2019) explore transfer learning in the field of medical imaging to address domain-specific difficulties. Different from these works that only consider the transferability of models after training, we investigate another dimension of model transferability in Section 6: the evolution of transferability during training with lrDecay.

## 3 COMMON BELIEFS IN EXPLAINING LRDECAY

### 3.1 GRADIENT DESCENT EXPLANATION

The practice of lrDecay in training neural networks dates back to LeCun et al. (1998). The most popular belief in the effect of lrDecay comes from the optimization analysis of Gradient Descent (GD) (LeCun et al., 1991). Although SGD is more practical in deep learning, researchers are usually satisfied with the analysis of GD considering that SGD is a stochastic variant of GD.

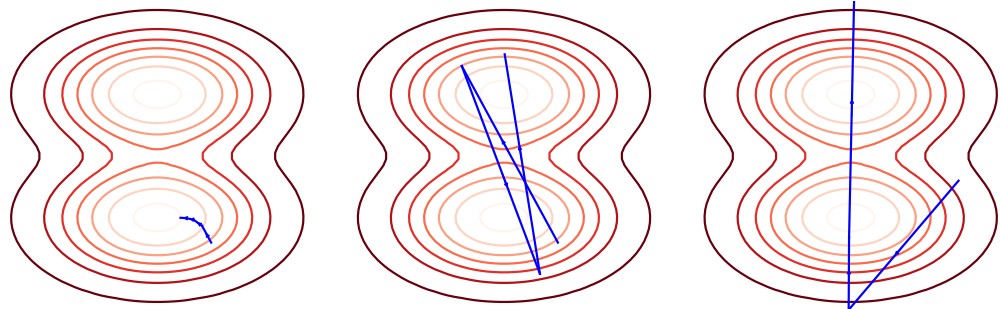

Figure 2: Gradient Descent explanation. From left to right: 1) learning rate is small enough to converge around a minimum, 2) moderate so that it bounces among minima, 3) too large to converge.

Specifically, LeCun et al. (1991) analyze the property of a *quadratic* loss surface which can be seen as a second-order approximation around a local minimum in nonconvex optimization. Learning rates are characterized by the relationship with eigenvalues of the Hessian at a local minimum. Denote $\eta$ the learning rate, $H$ the Hessian, $\lambda$ an eigenvalue of $H$, and $\mathbf{v}$ an eigenvector of $\lambda$. The behavior of the network along the direction $\mathbf{v}$ can be characterized as $(1 - \eta\lambda)^k \mathbf{v}$, with $k$ the iteration number. Convergence in the direction of $\mathbf{v}$ requires $0 < \eta < 2/\lambda$, while $\eta > 2/\lambda$ leads to divergence in the direction of $\mathbf{v}$. If $0 < \eta < 2/\lambda$ holds for every eigenvalue of the Hessian, the network will converge quickly (Figure 2 left). If it holds for some directions but not for all directions, the network will diverge in some directions and thus jump into the neighborhood of another local minimum (Figure 2 middle). If the learning rate is too large, the network will not converge (Figure 2 right). In particular, when oscillation happens, it means the learning rate is too large and should be decayed. The effect of lrDecay hence is to avoid oscillation and to obtain faster convergence. Note LeCun et al. (1991) only analyze a simple *one-layer* network. It may not hold for modern neural networks (see Section 4.1).

### 3.2 STOCHASTIC GRADIENT DESCENT EXPLANATION

Another common belief is the Stochastic Gradient Descent explanation, arguing that "with a high learning rate, the system is unable to settle down into deeper, but narrower parts of the loss function." [1] Although it is common, this argument has not been formally analyzed until very recently.

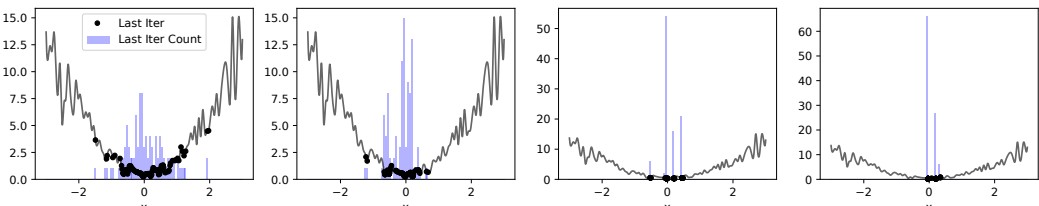

Figure 3: SGD explanation (taken from Kleinberg et al. (2018)). The first plot: an initially large learning rate helps escape spurious local minima. From the second to the fourth plots: after more rounds of learning rate decay, the probability of reaching the minimum becomes larger.

Under some assumptions, Kleinberg et al. (2018) prove SGD is equivalent to the convolution of loss surface, with the learning rate serving as the conceptual kernel size of the convolution. With an appropriate learning rate, spurious local minima can be smoothed out, thus helping neural networks escape bad local minima. The decay of learning rate later helps the network converge around the minimum. Figure 3 is an intuitive one-dimensional example. The first plot shows that a large learning rate helps escape bad local minima in both sides. The lrDecay in subsequent plots increases the probability of reaching the global minimum. Although intuitive, the explanation requires some assumptions that may not hold for modern neural networks (see Section 4.2).

---

[1] http://cs231n.github.io/neural-networks-3/#anneal

## 4    EXPERIMENTS AGAINST EXISTING EXPLANATIONS

Although the (Stochastic) Gradient Descent explanations in Section 3 account for the effect of lrDecay to some extent, in this section, we show by carefully-designed experiments that they are insufficient to explain the efficacy of lrDecay in modern neural networks. In all the experiments except for Section 6, we use a modern neural network named WideResNet (Zagoruyko & Komodakis, 2016). It is deep, wide, nonconvex, and suitable for datasets like CIFAR10 (Krizhevsky & Hinton, 2009).

### 4.1    EXPERIMENTS AGAINST THE GRADIENT DESCENT EXPLANATION

We train a WideResNet on CIFAR10 dataset with GD, decay the learning rate at different epochs, and report the training loss (optimization) as well as the test accuracy (generalization) in Figure 4. WideResNet and CIFAR10 are commonly used for studying deep learning (Zhang et al., 2017). CIFAR10 is not too large so that we can feed the whole dataset as a single batch using distributed training, computing the exact gradient rather than estimating it in mini-batches. Experiments show that lrDecay brings negligible benefit to either optimization or generalization. No matter when the learning rate is decayed, the final performances are almost the same. The instability in the beginning is related to the high loss wall described in Pascanu et al. (2013), which is not the focus of this paper.

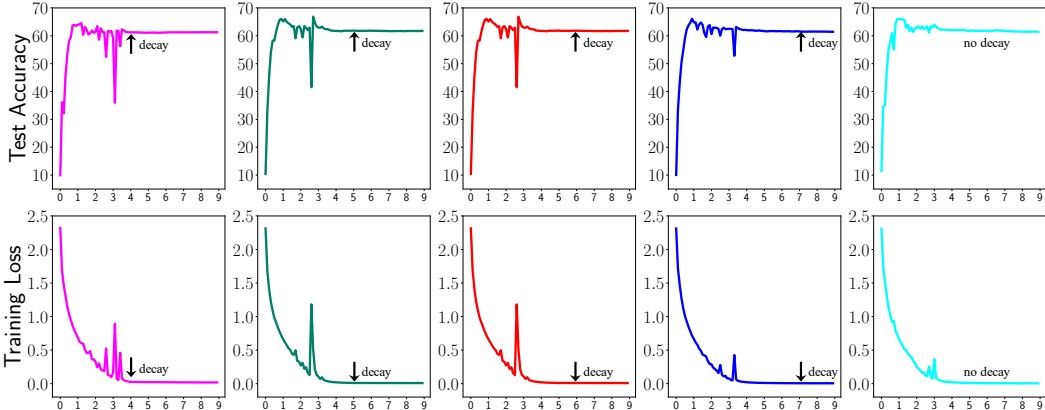

Figure 4: Training of WideResNet on CIFAR10 with Gradient Descent. X-axis indicates the number of epochs (in $10^3$). Arrows indicate the epoch with learning rate decay.

The above observation contradicts directly with the GD explanation in Section 3.1. The contradiction arises from the fact that LeCun et al. (1991) only analyze simple linear networks, and no wonder the explanation fails in modern non-linear deep networks. Recent studies (Keskar et al., 2017; Yao et al., 2018) reveal that large-batch training of modern networks can lead to very sharp local minima. Gradient Descent (the extreme of large batch training) can lead to even sharper local minima. In Figure 5, we calculate the largest ten eigenvalues[2] of the Hessian as well as the convergence interval ($0 < \eta < 2/\lambda$) for each eigenvalue for a trained WideResNet. The top eigenvalues reach the order of $\approx 200$. By contrast, eigenvalues of simple networks in LeCun et al. (1991) often lie in $[0, 10]$ (Figure 1 in their original paper). The spectrum of eigenvalues in modern networks is very different from that in simple networks analyzed by LeCun et al. (1991): the Hessian of modern networks has a much larger spectral norm.

The GD explanation in Section 3.1 attributes the effect of lrDecay to avoiding oscillation. Oscillation means there is a small divergence in some directions of the landscape so that the network bounces among nearby minima. However, the divergence factor $1 - \eta\lambda$ for the largest eigenvalue ($\approx 200$) is too large even for a small growth of learning rate. Thus, the learning rate is either small enough to converge in a local minimum or large enough to diverge. It is hardly possible to observe the oscillation in learning curves (Figure 2 middle), and diverging learning curves (Figure 2 right) can be discarded during hyperparameter tuning. Therefore, only stable solutions are observable where $\eta$ is small enough (Figure 2 left), leaving no necessity for learning rate decay. Indeed, when the

---

[2]Thanks to the advances of Xu et al. (2018); Yao et al. (2018), we can compute the eigenvalues directly.

learning rate is increased mildly, we immediately observe diverging learning curves (Section A.2). In short, the GD explanation cannot explain the effect of lrDecay in training modern neural networks.

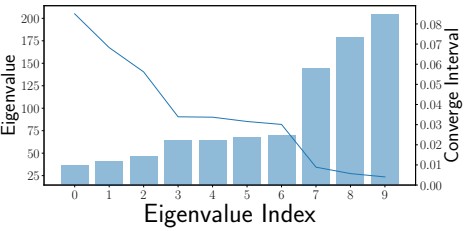

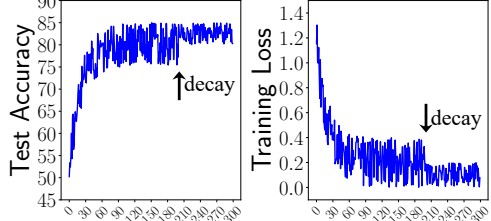

Figure 5: The largest ten eigenvalues $\lambda$ (blue curve) and converge intervals $(0, \frac{2}{\lambda})$ (bar) for WideResNet trained with Gradient Descent.

Figure 6: Expected behavior (but not observed) induced by the SGD explanation: best performances before and after decay are comparable.

## 4.2 EXPERIMENTS AGAINST THE STOCHASTIC GRADIENT DESCENT EXPLANATION

We follow the experiment setups in Section 4.1, but replace GD with SGD in Figure 7. According to the SGD explanation in Section 3.2, the effect of learning rate decay is to increase the probability of reaching a good minimum. If it is true, the model trained before decay can also reach minima, only by a smaller probability compared to the model after decay. In other words, the SGD explanation indicates the best performances before and after decay are the same. It predicts learning curves like Figure 6. However, Figure 7 does not comply with the SGD explanation: the best performances before and after lrDecay are different by a noticeable margin. Without lrDecay (the rightmost column in Figure 7), the performance plateaus and oscillates, with no chance reaching the performance of the other columns after decay. The performance boost after learning rate decay is widely observed (Figure 1(b) for example). However, possibly due to the violation of its assumptions (Kleinberg et al., 2018), the SGD explanation cannot explain the underlying effect of lrDecay.

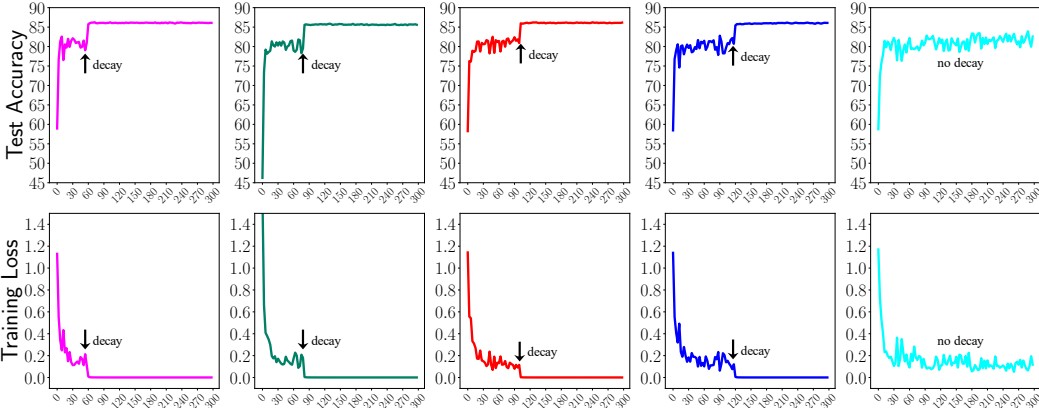

Figure 7: Training of WideResNet on CIFAR10 with SGD. X-axis indicates the number of epochs. Arrows show the moment of learning rate decay. The rightmost plots show results without decay.

## 5 AN EXPLANATION FROM THE VIEW OF PATTERN COMPLEXITY

Section 4 uncovers the insufficiency of common beliefs in explaining lrDecay. We thus set off to find a better explanation. Mangalam & Prabhu (2019); Nakkiran et al. (2019) reveal that SGD (without learning rate decay) learns from easy to complex. As learning rates often change from large to small in typical learning rate strategies, we hypothesize that the complexity of learned patterns is related to the magnitude of learning rates. Based on this, we provide a novel explanation from the view of pattern complexity: **the effect of learning rate decay is to improve the learning of complex patterns while the effect of an initially large learning rate is to avoid memorization of noisy**

**data**. To justify this explanation, we carefully construct a dataset with tractable pattern complexity, and record model accuracies in simple and complex patterns separately with and without lrDecay.

### 5.1 PATTERN SEPARATION 10 (PS10) DATASET WITH TRACTABLE PATTERN COMPLEXITY

The explanation we propose involves pattern complexity, which is generally conceptual and sometimes measured with the help of a simple auxiliary model as in Mangalam & Prabhu (2019); Nakkiran et al. (2019). Here we try to formalize the idea of pattern complexity: the complexity of a dataset is defined as the expected class conditional entropy: $C(\{(x_i, y_i)\}_{i=1}^n) = \mathbb{E}_y H(P(x|y))$, where $H$ denotes the entropy functional. The complexity of patterns depends on the complexity of the dataset they belong to. Higher $C$ means larger complexity because there are averagely more patterns in each class to be recognized (consider an animal dataset with 10 subspecies in each species vs. an animal dataset with 100 subspecies in each species).

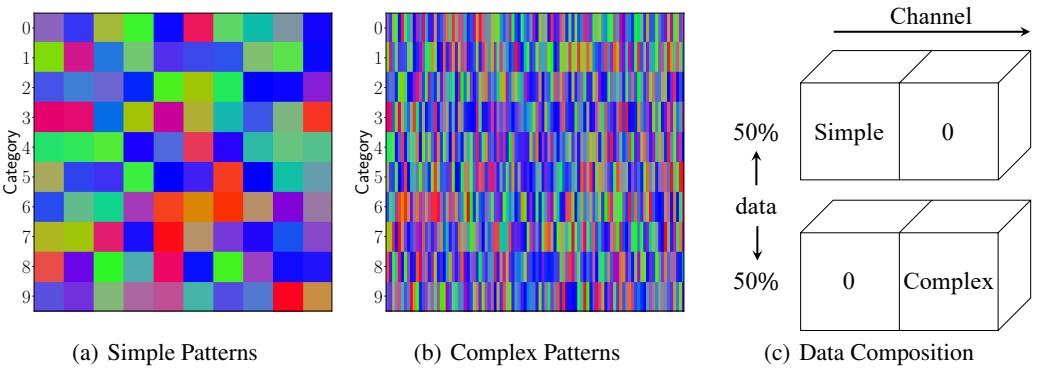

(a) Simple Patterns      (b) Complex Patterns      (c) Data Composition

Figure 8: The PS10 dataset. (a) Simple patterns: 10 patterns per category, complexity $\log_2 10$. (b) Complex patterns: 100 patterns per category, complexity $\log_2 100$. (c) Data composition: half of the data only contain simple patterns while another half only contain complex patterns.

Equipped with the formal definition of complexity, we construct a **Pattern Separation 10 (PS10)** dataset with ten categories and explicitly separated simple patterns and complex patterns. We first generate a simple sub-dataset together with a complex sub-dataset in $\mathbb{R}^3$. As shown in Figure 8(a) and Figure 8(b), patterns are visualized as colors because they lie in $\mathbb{R}^3$. The category label can be identified by either simple patterns or complex patterns. We then merge the two sub-datasets into one dataset. The merging method in Figure 8(c) is specifically designed such that the simple subset and complex subset are fed into different channels of the WideResNet. This mimics the intuition of patterns as the eye pattern and the nose pattern have different locations in an image of human face. To be compatible with the sliding window fashion of convolutional computation, we make patterns the same across spatial dimensions of height and weight to have the same image size as CIFAR10.

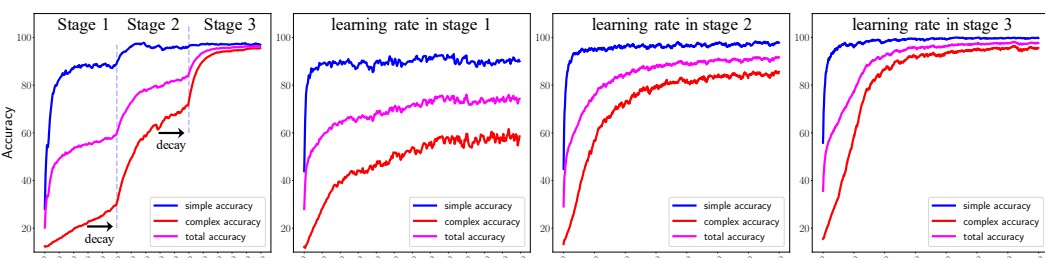

Figure 9: Experiments with lrDecay and without lrDecay (constant learning rates) w.r.t accuracies in different patterns. From left to right: Train with lrDecay; Train with a constant learning rate equal to the learning rate in Stage 1, 2, and 3 of lrDecay, respectively. The X-axis shows the epoch number.

## 5.2 THE EFFECT OF DECAY: IMPROVE LEARNING OF MORE COMPLEX PATTERNS

To reveal the effect of decaying the learning rate, we compare experiments with and without lrDecay. For those without lrDecay, we set the learning rates equal to the learning rate of each stage in lrDecay. We measure not only the total accuracy but also the accuracies on simple and complex patterns separately. These accuracies are plotted in Figure 9.

The first plot in Figure 9 clearly shows the model first learns simple patterns quickly. The boost in total accuracy mainly comes from the accuracy gain on complex patterns when the learning rate is decayed. Plots 2, 3, and 4 show the network learns more complex patterns with a smaller learning rate, leading to the conclusion that learning rate decay helps the network learn complex patterns.

## 5.3 THE EFFECT OF AN INITIALLY LARGE LEARNING RATE: AVOID FITTING NOISY DATA

Figure 9 seems to indicate that an initially large learning rate does nothing more than accelerating training: in Plot 4, a small constant learning rate can achieve roughly the same accuracy compared with lrDecay. However, by adding 10% noisy data to mimic real-world datasets, we observe something interesting. Figure 10 shows the accuracies on simple patterns, complex patterns, and noise data when we add noise into the dataset. Plot 2 in Figure 10 shows an initially large learning rate helps the accuracy on complex patterns. Plot 3 in Figure 10 further shows the accuracy gain on complex patterns comes from the suppression of fitting noisy data. (Note that a larger accuracy on noisy data implies overfitting the noisy data, which is undesirable.) In short, the memorizing noisy data hurts the learning of complex patterns but can be suppressed by an initially large learning rate.

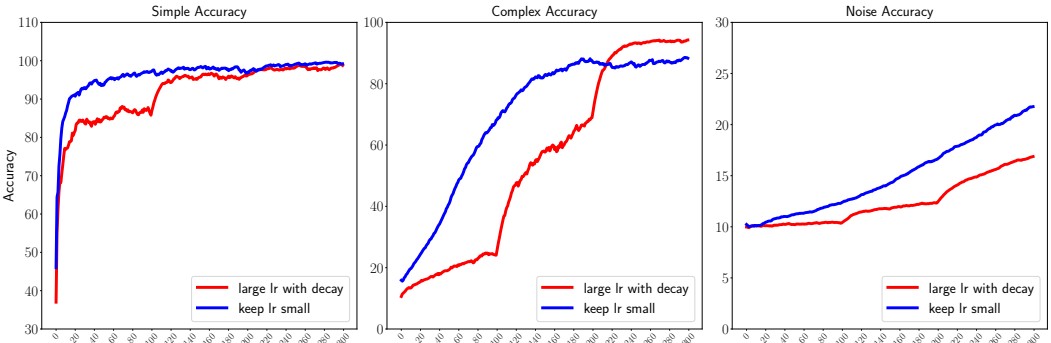

Figure 10: Comparison between lrDecay and a constant small learning rate on the PS10 dataset with 10% noise. Accuracies on simple patterns, complex patterns, and noise data are plotted respectively.

Empirically, Li et al. (2019) report that an initially large learning rate with decay outperforms a small and constant learning rate. They suspect that the network starting with an initially small learning rate will be stuck at some spurious local minima. Our experiments provide an alternative view that spurious local minima may stem from noisy data. And the regularization effect of an initially large learning rate is to suppress the memorization of noisy data.

## 6 THE IMPLICATION OF lrDecay ON MODEL TRANSFERABILITY

Section 5 examines the proposed explanation on the PS10 dataset. Now we further validate the explanation on real-world datasets. Because there are no clearly separated simple and complex patterns in real-world datasets, it is difficult to directly validate the explanation. The proposed explanation suggests that SGD with lrDecay learns patterns of increasing complexity. Intuitively, more complex patterns are less transferable, harder to generalize across datasets. Thus an immediate implication is that SGD with lrDecay learns patterns of decreasing transferability. We validate it by transfer-learning experiments on real-world datasets, to implicitly support the proposed explanation.

The transferability is measured by transferring a model from ImageNet to different target datasets. To get models in different training stages, we train a ResNet-50 on ImageNet from scratch and save checkpoints of models in different stages. The learning rate is decayed twice, leading to three

stages. Target datasets for transferring are: (1) Caltech256 (Griffin et al., 2007) with 256 general object classes; (2) CUB-200 (Wah et al., 2011) with 200 bird classes; (3) MITIndoors (Quattoni & Torralba, 2009) with 67 indoor scenes; (4) Sketch250 (Eitz et al., 2012) with sketch painting in 250 general classes. Sketch250 is the most dissimilar to ImageNet because it contains sketch paintings.

We study two widely-used strategies of transfer learning: "fix" (ImageNet snapshot models are only used as fixed feature extractors) and "finetune" (feature extractors are jointly trained together with task-specific layers). Let $acc_i$ denotes the accuracy of stage $i$ snapshot model on ImageNet and $tacc_i$ denotes the accuracy of transferring the snapshot to the target dataset, then the transferability of *additional* patterns learned in stage $i$ is defined as $\frac{tacc_i - tacc_{i-1}}{acc_i - acc_{i-1}}, i = 2, 3$. By definition, the transferability of patterns from ImageNet to ImageNet is 1.0, complying with common sense. The transferability is plotted in Figure 11. Table 2 contains the accuracies used to compute it.

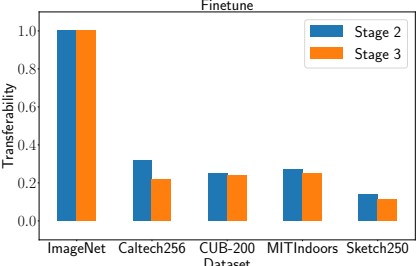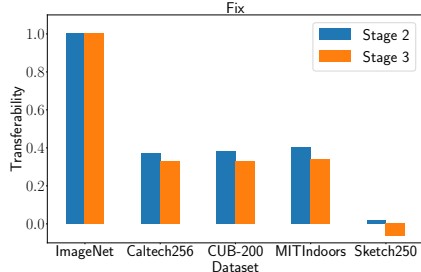

Figure 11: Transferability of additional patterns learned in each stage w.r.t different target datasets.

In all experiments, we find that the transferability of additional patterns learned in stage 3 is less than that in stage 2. Besides, in Sketch250 dataset, the transferability of additional patterns learned in stage 3 is negative. These findings support our claim that additional patterns learned in later stages of lrDecay are more complex and thus less transferable. They also suggest deep model-zoo developer provide pre-trained model snapshots in different stages so that downstream users can select the most transferable snapshot model according to their tasks.

## 7 CONCLUSION

In this paper, we dive into how learning rate decay (lrDecay) helps modern neural networks. We uncover the insufficiency of common beliefs and propose a novel explanation: the effect of decaying learning rate is to improve the learning of complex patterns, and the effect of an initially large learning rate is to avoid memorization of noisy data. It is supported by experiments on a dataset with tractable pattern complexity as well as on real-world datasets. It would be interesting to further bridge the proposed explanation and the formal analysis of optimization procedure.

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

# A APPENDIX

## A.1 AUTODECAY

Experiments in Section 5.2 implies that not all complex patterns are learnable under a constant learning rate. The training under a certain learning rate has no effect when the loss plateaus. This indicates we can expedite the training process by killing the over-training of each stage (decay the learning rate when the loss plateaus) with little influence on the performance. To validate the implication, we propose AutoDecay to shorten the useless training and check if the performance of the model can be untouched. In Figure 7, it appears obvious to decide the optimal moment to decay when we have a big picture of the training process. The problem is, however, how can we make a decision to decay depending on the current and past observations. It is a non-trivial problem given that the statistics exhibit noticeable noise.

### A.1.1 PROBLEM FORMULATION

We formalize the observed training loss into two parts: $\hat{\ell}(t) = \ell(t) + \epsilon(t)$, with $\ell(t)$ the ground truth loss (unobservable) and $\epsilon(t)$ the noise introduced by SGD. Here $t$ indicates the training process (typically the epoch number) and takes value in $\mathcal{N} = \{1, 2, 3, \ldots\}$. To simplify the problem, we assume $\epsilon(t)$ is independent with $t$ and $\epsilon(t)$ is independent of $\epsilon(t')(t' \neq t)$ in SGD. The nature of noise gives rise to the zero-expectation property $\mathbb{E}\,\epsilon(t) = 0$. Denote $\sigma^2 = \text{Var}\,\epsilon(t)$ the variance of the noise. Due to the noise of SGD, the observed training loss usually vibrates in a short time window but decreases in a long time window. Our task is to find out whether the loss value is stable in the presence of noise.

### A.1.2 PROBLEM SOLUTION

**Exponential Decay Moving Average (EDMA) with Bias Correction.** Observations with lower variance are more trustworthy. However, there is nothing we can do about the variance of $\hat{\ell}(t)$. We consider computing a low-variance statistic about $\hat{\ell}(t)$. We adopt moving average with bias correction(Kingma & Ba, 2015). Let $g(t)$ be the moving average of $\ell(t)$ and $\hat{g}(t)$ be the moving average of $\hat{\ell}(t)$. The explicit form is in Equation 1, where $\beta \in (0, 1)$ is the decay factor in EDMA.

$$
\begin{aligned}
g(t) &= \frac{\sum_{i=0}^{t-1} \beta^i \ell(t-i)}{\sum_{i=0}^{t-1} \beta^i} = \frac{1-\beta}{1-\beta^t} \sum_{i=0}^{t-1} \beta^i \ell(t-i), t \geq 1 \\
\hat{g}(t) &= \frac{\sum_{i=0}^{t-1} \beta^i \hat{\ell}(t-i)}{\sum_{i=0}^{t-1} \beta^i} = \frac{1-\beta}{1-\beta^t} \sum_{i=0}^{t-1} \beta^i \hat{\ell}(t-i), t \geq 1
\end{aligned}
\tag{1}
$$

The recursive (and thus implicit) form is in Equation 2. It enables us to compute the statistic $\hat{g}$ online (without storing all the previous $\{\hat{\ell}(i)|i < t\}$) at the cost of maintaining $\hat{f}(t)$.

$$
\begin{aligned}
f(0) &= 0, f(t) = \beta f(t-1) + (1-\beta)\ell(t) \implies g(t) = \frac{f(t)}{1-\beta^t}(t \geq 1) \\
\hat{f}(0) &= 0, \hat{f}(t) = \beta \hat{f}(t-1) + (1-\beta)\hat{\ell}(t) \implies \hat{g}(t) = \frac{\hat{f}(t)}{1-\beta^t}(t \geq 1)
\end{aligned}
\tag{2}
$$

As $\hat{g}(t)$ is a linear combination of $\{\hat{\ell}(i)|i \leq t\}$, it is easy to show $\hat{g}(t)$ is unbiased:

$$
\mathbb{E}\hat{g}(t) = \mathbb{E}\frac{\sum_{i=0}^{t-1} \beta^i \hat{\ell}(t-i)}{\sum_{i=0}^{t-1} \beta^i} = \frac{\sum_{i=0}^{t-1} \beta^i \mathbb{E}\hat{\ell}(t-i)}{\sum_{i=0}^{t-1} \beta^i} = \frac{\sum_{i=0}^{t-1} \beta^i \ell(t-i)}{\sum_{i=0}^{t-1} \beta^i} = g(t)
$$

The variance of $\hat{g}(t)$ is

$$\text{Var}\hat{g}(t) = \frac{(1-\beta)^2}{(1-\beta^t)^2}\text{Var}\sum_{i=0}^{t-1}\beta^i\hat{\ell}(t-i) = \frac{(1-\beta)^2\sigma^2}{(1-\beta^t)^2}\sum_{i=0}^{t-1}\beta^{2i} = \frac{1-\beta}{1+\beta}\frac{1+\beta^t}{1-\beta^t}\sigma^2 \quad (3)$$

The fact that $\beta \in (0,1)$ indicates $\text{Var}\hat{g}(t)$ is monotonically decreasing. Typically $\beta = 0.9$ (Figure 13), and the variance can rapidly converge to $0.05\sigma^2$, much smaller than the variance of the noise. $\hat{g}(t)$ well represents the unobservable $g(t)$. If $\ell(t)$ gets stable, we shall observe that $\hat{g}(t)$ is stable, too.

**Criterion of Being Stable**. We only want to decay the learning rate when the loss plateaus, i.e. when the loss is stable. For observed values of $G = \{\hat{g}(i)|i - W + 1 \le i \le t\}$ within the window size of $W$, we call them stable if $\frac{\max G - \min G}{\min G + \epsilon} < \eta$, where $\epsilon$ is a small constant that prevents zero-division error, and $\eta$ indicates the tolerance of variation.

**Criterion of Significant Drop**. When we keep decaying the learning rate, there comes a time when the learning rate is too small and the network cannot make any progress. When it happens, we should terminate the training. Termination is adopted when their is no significant drop between the stable value and the original value $\hat{g}(0)$. To be specific, the criterion of significant drop is $\frac{\hat{g}(t)+\epsilon}{\hat{g}(0)+\epsilon} \le \zeta$, where $\epsilon$ is a small constant that prevents zero-division error, and $\zeta$ indicates the degree of drop.

The entire procedure of AutoDecay is described in Figure 12.

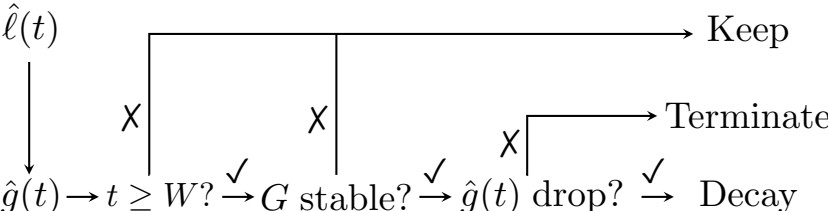

Figure 12: Decision Procedure of AutoDecay. The counter $t$ is reset to 0 at the action of "Decay".

### A.1.3 EXPERIMENTS

We try AutoDecay on ImageNet (Russakovsky et al., 2015) to test whether it can expedite the training without hurting the performance. We are not trying to set up a new state-of-the-art record. We train a ResNet-50 model on ImageNet following the official code of PyTorch. The only change is we replace the StepDecay strategy with the proposed AutoDecay strategy. Each experiment costs roughly two days with 8 TITAN X GPUs. The results in Table 14 show that AutoDecay can shorten the training time by 10% without hurting the performance (even bringing a slight improvement), successfully vaidates the proposed explanation in this paper.

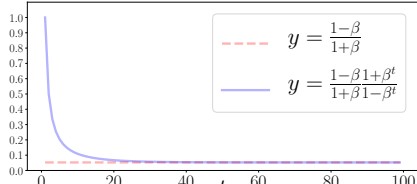

Figure 13: Variance reduction when $\beta = 0.9$

| Method | ImageNet | | |
|---|---|---|---|
| | epochs | top1 | top5 |
| **StepDecay** | 90 | 75.80 | 92.76 |
| **AutoDecay** | 81 | 75.91 | 92.81 |

Figure 14: Results of AutoDecay.

### A.2 LARGER LR LEADS TO DIVERGENCE IN GD FOR MODERN NEURAL NETWORKS

When we increase the learning rate mildly for Gradient Descent, we immediately observe diverging learning curves (Figure 15), which echos with the reason mentioned in Section 4.1 why the Gradient Descent explanation fails to work in modern neural networks: modern neural networks have a very

large spectrum norm at a local minimum, and even a small growth of learning rate can lead to divergence. In other words, training modern neural networks with GD must use a small enough learning rate, dismissing the value of learning rate decay.

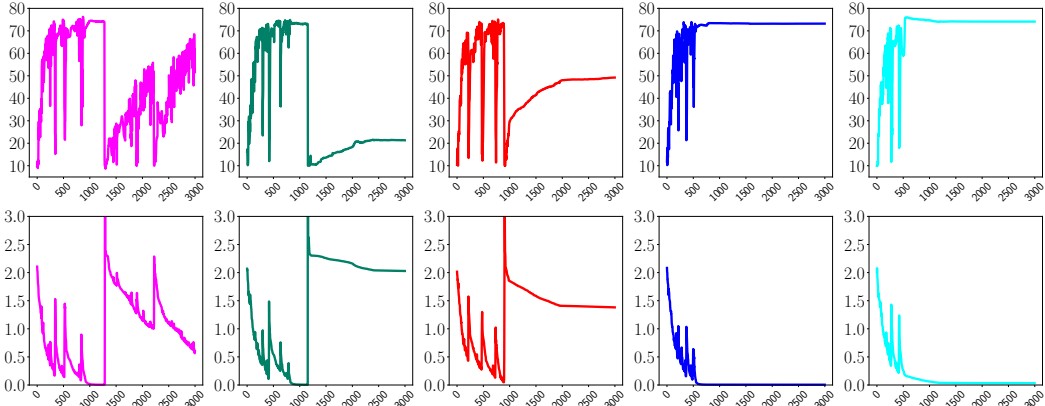

Figure 15: Training of WideResNet on CIFAR10 by Gradient Descent with a mildly larger learning rate. X-axis indicates number of epochs. Arrows and texts show the moment of learning rate decay.

### A.3 ACCURACIES TO COMPUTE THE TRANSFERABILITY IN SECTION 6

| Dataset | Finetune | | | | | Fix | | | | |
|---|---|---|---|---|---|---|---|---|---|---|
| | stage1 | stage2 | **stage2** | stage3 | **stage3** | stage1 | stage2 | **stage2** | stage3 | **stage3** |
| ImageNet | 54.01 | 69.33 | **1.00** | 75.91 | **1.00** | 54.01 | 69.33 | **1.00** | 75.91 | **1.00** |
| Caltect256 | 77.87 | 82.73 | **0.32** | 84.20 | **0.22** | 74.10 | 79.79 | **0.37** | 81.95 | **0.33** |
| CUB_200 | 75.89 | 79.74 | **0.25** | 81.34 | **0.24** | 58.11 | 63.95 | **0.38** | 66.09 | **0.33** |
| MITIndoors | 72.91 | 77.09 | **0.27** | 78.73 | **0.25** | 61.34 | 67.54 | **0.40** | 69.78 | **0.34** |
| Sketch250 | 77.80 | 79.92 | **0.14** | 80.64 | **0.11** | 64.34 | 64.58 | **0.02** | 64.18 | **-0.06** |

Table 2: Accuracy and transferability of ImageNet models in different stages. Normal values indicate accuracy and bold values indicate transferability.

