# OpenReview forum: "How Does Learning Rate Decay Help Modern Neural Networks?"
_ICLR.cc/2020/Conference — Reject_

### Official Review · AnonReviewer2 · 2019-10-22
**Official Blind Review #2**

**Rating:** 3

**Review:**

The paper investigates the role of learning rate decay in neural network training. While there are prevalent ideas of how/why learning rate decay help both optimization and generalization of neural networks, this work proposes interpretation based on pattern complexity. The mechanism the paper proposes is that initial learning rate helps ignore noise in the beginning and decayed learning rate help to learn complex patterns.

The question the paper tackles is a very important question in understanding deep learning that requires careful study. As the learning rate schedule benefits neural network models beyond specific domain, this question has high significance and potential impact.

The authors propose a view from a pattern complexity. I think this view point is well motivated in the light of Li et al. (2019). The artificial task constructed by authors reveals that with different phases of learning rates, different types of complexity are being learnt. Also transfer learning tasks also reveals that one might want to use models before dropping the learning rate for purpose of transfer learning.

While the question studied is of high importance, I am not confident that the methods presented justifies the claims in the paper. Especially authors analysis on how previous understanding of learning rate decay is flawed doesn’t seem fully supported. With current submission I slightly lean towards rejecting.

One reason is that overall the details of experiments are not specific enough that I personally wouldn’t feel comfortable reproducing the results. For example, in Section 4, was data augmentation or weight decay used? What was the mini-batch size used for SGD experiments? Which learning rate is used for  large learning rate and to which learning rate was it decayed to? I think even one is using full batch GD, the performance shown in Figure 4 which is just above 60% test accuracy is quite low for models like WideResNet.

Also I do not agree with the claim that SGD explanation leads to training curves as in Figure 6. The plot indicates large learning rate / lower learning rate have about the same probability of reaching the 'global minima’. However as described in section 3.2, the role of lowering the learning rate should be increasing the probability of global minima. To me experiments shown in Figure 7, supports both explanations from Lecun et al (1991) Kleinberg et al. (2018) described in Table1.

Few extra comments:

One question I have regarding transferability analysis. Kornblith et al. (2019) showed that better Imagenet models transfers better. However section 6 shows that  transferability is higher for stage 2 which has higher error. How do you reconcile this discrepancy?

I don't quite comprehend the motivation for putting two complexity patterns in disjoint channels. Especially the comment “This mimics the intuition of patterns as the eye pattern and the nose pattern have different locations in an image of human face.” is unclear. Could you elaborate?

Figure 15 seems incomplete. Should there be arrows for decay points? Please also include details of those experiments too.


**Experience Assessment:**

I have read many papers in this area.

**Review Assessment: Checking Correctness Of Derivations And Theory:**

N/A

**Review Assessment: Checking Correctness Of Experiments:**

I carefully checked the experiments.

**Review Assessment: Thoroughness In Paper Reading:**

I read the paper at least twice and used my best judgement in assessing the paper.

---

### Official Review · AnonReviewer3 · 2019-10-22
**Official Blind Review #3**

**Rating:** 1

**Review:**

Contributions:
This paper investigates the use of learning rate decay in deep neural networks.  The main contribution is an empirical analysis trying to understand lr decay. Authors claim that the high-learning rate phase provides regularization and prevents the network to fit/memorize noisy data initially to focus on simple patterns.

Authors design a set of experiments showing that lr decay allows SGD to first fit ‘simple patterns’ instead of more complex/noisy one. They also show that the model transferability decreases through training.

In addition to those experiments, authors provide experimental results which aims at contradicting common beliefs on lr decay.

Novelty/Significance:
My main concern is about the novelty/significance of the paper.  Similar argument has already been made in (Nakiran et al., 2019) or (Li et al. 2019), which are cited in the paper, and other works such as “On the Spectral Bias of Neural Networks”. In addition, prior to this work, the paper “Three factors Influencing Minima in SGD” empirically showed that the noise in SGD (which is controlled by the learning rate) prevents memorization (see Figure 6 in their paper). Although the latter work did not focus the lr decay, it is not clear to me how the main argument of the current paper differs from it.

Additional comments:
-	It is unclear to me why the model the model before and after decay should have the same training performance in section 4.2. This would assume a specific geometry of the loss function (i.e. that the loss surface is not significantly narrower so that you can reach it with a high-learning rate), but authors do not provide evidence of it.
-	It would be nice to provide more details about the experimental settings. What is the base learning rate? Did you try different learning rate? Do you use with SGD-momentum, weight-decay?
-	How to combine the current explanation of lr decay with the observation that learning rate warm-up has been a successful way to train neural network in the large batch setting?
-	The transferability experiment confounds two factors: the number of iteration steps   and the learning rates. Does training a model with low learning rate, but with a number a step equal to high learning rate lead to lower transferability?
-	It would be nice to validate the hypothesis on more datasets/models.



**Experience Assessment:**

I have published one or two papers in this area.

**Review Assessment: Checking Correctness Of Derivations And Theory:**

I assessed the sensibility of the derivations and theory.

**Review Assessment: Checking Correctness Of Experiments:**

I carefully checked the experiments.

**Review Assessment: Thoroughness In Paper Reading:**

I read the paper thoroughly.

---

### Official Review · AnonReviewer1 · 2019-10-28
**Official Blind Review #1**

**Rating:** 1

**Review:**

Summary: This paper investigates the way decaying the learning rate helps the training of neural networks. First the paper discusses about other existing hypothesis such as the “Gradient Descent Hypothesis” by Lecun et al 1991 and SGD explanation by Kleinberg et al 2018. Then the paper tries to find contradicting examples against those two hypothesis with experiments. Then they propose their explanation which suggests that initially fitting noisy data and then decaying it helps it to learn more complex data. Then the paper tries to experimentally explain why the other explanations fail and theirs is better.

This paper is very badly written. The authors should definitely rethink about the organization of the paper. The first pages is mostly about the background material with figures taken from other papers. The terminology that is being used in this paper is very vague. They used the term complex patterns in the paper, but don’t even explain it until page 6. When the paper explains it, still the notion itself how to compute it in a tractable way is a bit vague for neural networks.

This paper proposes all those different explanations of how SGD works just as if they are completely orthogonal. However, for example learning both the proposed explanation in this paper and the fact that learning rate decay improves stability can both be true.

The experimental arguments are quite vague in this paper. The authors should give more details about their experimental setup, for example what is the starting learning rate, ending learning rate, details of scheduling, what type of distributed training method (sync or async SGD?) and etc… is missing. Without those details it is very difficult to interpret the experimental conclusions in for example section 4.1. In Figure 5, what does the converge interval mean? In Figure 7 and 4 all the curves are so similar, why so?
The arguments that learning rate decaying helps models to learn complex patterns and without decay the models can not learn those is a very strong claim and not very well supported in this paper.

The details of how the PS10 dataset is constructed is somewhat vague. In terms of arguments that decaying learning rates help the neural networks to learn simpler examples first and the harder ones have already been done in other papers such as Li et al 2019 which is also cited in this paper. It is not completely clear what this paper contributes over those other existing papers :)


Questions:
How does the cyclic learning rates come into the picture. Can one explain why cyclic learning rates works with the hypothesis that this paper processes?
How would different types of annealing methods come into the picture: e.g. linearly decaying the learning rate, exponential decay and etc…
Why are there two columns of “supported” in Table 1? What is the difference between them.
How does the adaptive learning rate algorithms such as Adam come into the picture with the learning rate decay hypothesis?

**Experience Assessment:**

I have published one or two papers in this area.

**Review Assessment: Checking Correctness Of Derivations And Theory:**

I carefully checked the derivations and theory.

**Review Assessment: Checking Correctness Of Experiments:**

I assessed the sensibility of the experiments.

**Review Assessment: Thoroughness In Paper Reading:**

I read the paper at least twice and used my best judgement in assessing the paper.

---

### Decision · Program_Chairs · 2019-12-19

**Decision:**

Reject

**Comment:**

This paper seeks to understand the effect of learning rate decay in neural net training. This is an important question in the field and this paper also proposes to show why previous explanations were not correct. However, the reviewers found that the paper did not explain the experimental setup enough to be reproducible. Furthermore, there are significant problems with the novelty of the work due to its overlap with works such as (Nakiran et al., 2019), (Li et al. 2019) or (Jastrzębski et al. 2017).